

# Measurement report: Vanadium-containing ship exhaust particles detected in and above the marine boundary layer in the remote atmosphere

Maya Abou-Ghanem[1], Daniel M. Murphy[1], Gregory P. Schill[1], Michael J. Lawler[1,2], and Karl D. Froyd[1,2]

[1]Chemical Sciences Laboratory, National Oceanic and Atmospheric Administration, Boulder, 80305, USA
[2]Cooperative Institute for Research in Environmental Sciences, University of Colorado Boulder, Boulder, 80309, USA

*Correspondence to*: Maya Abou-Ghanem (maya.abou-ghanem@tofwerk.com)

**Abstract.** Each year, commercial ships emit over 1.2 Tg of particulate matter (PM) pollution into the atmosphere. These ships rely on the combustion of heavy fuel oil, which contains high levels of sulfur, large aromatic organic compounds, and metals. Vanadium is one of the metals most commonly associated with heavy fuel oil and is often used as a tracer for PM from ship exhaust. Previous studies have suggested that vanadium-containing PM has impacts on human health and climate due to its toxicological and cloud formation properties, respectively; however, its distribution in the atmosphere is not fully understood, which limits our ability to quantify the environmental implications of PM emitted by ships. Here, we present data obtained from Particle Analysis by Laser Mass Spectrometry (PALMS) instrument on the NASA DC-8 aircraft during the 2016–2018 Atmospheric Tomography Mission (ATom) and show that ~1% of the accumulation mode particles measured in the marine boundary layer of the central Pacific and Atlantic Ocean contain vanadium. These measurements, which were made without targeting ship plumes, suggest that PM emitted by ships is widespread in the atmosphere. Furthermore, we observed vanadium-containing ship exhaust particles at altitudes up to 13 km, which demonstrates that not all ship exhaust particles are immediately removed via wet deposition processes. In addition, using laboratory calibrations, we determined that most vanadium-containing ship exhaust particles can contain up to a few wt % vanadium. This study furthers our understanding of both the chemical composition and distribution of PM emitted by ships, which will allow us to better constrain the climate, health, and air quality implications of these particle types in the future.

## 1 Introduction

The world economy is highly reliant on the shipping industry, which is responsible for facilitating 80% of global trade (UNCTAD, 2022). The strong demand for the import and export of goods supports approximately 50,000 active ships worldwide (UNCTAD, 2018). Traditionally, these fleets have relied on the combustion of heavy fuel oil for vessel propulsion, which is the residue obtained from petroleum distillation. These low-cost fuels are rich in aromatic hydrocarbons, sulfur, and heavy metals; as a result, emissions from marine vessels have a significant contribution to global air pollution, with estimated



emissions of 21 and 12 Tg of $NO_x$ and $SO_x$, respectively (Eyring et al., 2005). Additionally, the combustion of heavy fuel oil

from ships contributes to approximately 1.67 Tg $y^{-1}$ of particulate matter with an aerodynamic diameter of 10 µm or smaller

($PM_{10}$). These particulates emitted by ships can mix with marine aerosol and influence the formation of clouds (Hobbs et al.,

2000), which can have climate impacts by altering Earth's radiative budget (Lauer et al., 2007). Furthermore, it is estimated

that the inhalation of PM emitted from ships contributes to 60,000 human deaths per year (Corbett et al., 2007). One toxic

component of this PM is heavy metals, which have been previously linked to cardiovascular damage (Ye et al., 2018) and lung

cancer (Bollati et al., 2010).

Many analytical techniques have characterized heavy metals in PM emitted by ships including, inductively coupled plasma

mass spectrometry (Moreno et al., 2010; Celo et al., 2015) and X-ray fluorescence (Moldanová et al., 2013); however, these

bulk analysis techniques have low temporal and spatial resolution because they require samples to be collected onto filters

prior to analysis. To overcome these limitations, single-particle mass spectrometry has been used for real-time chemical

analysis of PM from ships. This technique is advantageous because it characterizes both the non-refractory (e.g., organic,

nitrate, and sulfate) and refractory (e.g., heavy metals) components of individual particles emitted from ships, which can

provide information on particle mixing state (i.e., internally or externally mixed). For example, previous single-particle mass

spectrometry studies have classified ship-borne PM as internal mixtures based on unique chemical signatures in the mass

spectra, which include vanadium, nickel, iron, organics, nitrate, ammonium, and sulfate (Ault et al., 2010; Passig et al., 2021;

Froyd et al., 2019; Healy et al., 2009).

Over the past several decades, vanadium and nickel have been used as chemical tracers for ship exhaust particles in the marine

atmosphere (Viana et al., 2009; Zhang et al., 2014). To our best knowledge, these chemical signatures have only been used for

the identification of ship exhaust particles and not for the quantification of heavy metals in single particles. In addition, the

majority of single-particle mass spectrometry measurements of PM from ships have been conducted at port or within ship

plumes, which limits our ability to understand how the chemical composition of ship PM evolves as it undergoes atmospheric

aging.

From 2016-2018, global in-situ aircraft-based aerosol measurements were made using the particle analysis by laser mass

spectrometry (PALMS) instrument, which was on board the NASA DC-8 aircraft during the Atmospheric Tomography

Mission (ATom). Analysis of the comprehensive PALMS datasets obtained during ATom reveal that ~1% of the accumulation

mode particles in the marine boundary layer (MBL) resembled that of PM emitted from ships due to the presence of vanadium,

nickel, iron, sulfate, and organic material. These measurements, which were made without specifically targeting ship plumes,

suggest that PM from ships can remain in the atmosphere for extended periods of time. In this work, we present the application

of laboratory calibrations to the ATom data set to quantify vanadium in individual ship particles. In addition, we also

investigate how the chemical composition of these particles changes as they are processed in the atmosphere.

The measurements presented in this study are prior to the 2020 International Maritime Organization (IMO) sulfur regulation,

which was aimed to reduce the air pollution burden of marine vessels by mandating the use of cleaner fuels. Specifically, this

policy required the global shipping industry to transition from 3.5 to 0.5 wt % sulfur fuel (IMO 2020 – cutting sulphur oxide



emissions, 2023). Even stricter sulfur regulations (0.1 wt %) have been implemented in emission control areas (ECAs), which
have been implemented near certain coastal regions in the US, Canada, Europe, and China to protect populations residing near
highly trafficked shipping areas. The sulfur targets in the 2020 IMO regulation and ECAs can be achieved through the addition
of scrubbers to capture $SO_2$ emitted during heavy-fuel oil combustion or by using cleaner fuel alternatives (e.g., distillates or
natural gas) (Lehtoranta et al., 2019). These methods can also result in a reduction in the total number of vanadium-containing
particles emitted by ships (Yu et al., 2021; Xiong et al., 2023); however, the reductions are not complete as vanadium-
containing ship exhaust particles are still detected in ECAs (Passig et al., 2021; Xiong et al., 2023). Although this study was
conducted before the 2020 IMO policy, it provides detailed baseline compositional and geographical measurements for
vanadium-containing ship exhaust particles in the remote atmosphere that can be used to compare the changes in distribution,
chemical composition, and atmospheric aging of this particle type after the enactment of the sulfur policy.

## 2 Methods

### 2.1 Single-particle measurements during ATom using PALMS


The ATom dataset is especially valuable for understanding the prevalence of ship particles because the globe-spanning flights
were largely predetermined and made no effort to target or avoid ship plumes or shipping lanes. Although this sampling method
results in a loss of detail about fresh ship exhaust particles, it provides fairly unbiased measurements for these particles and
allows for a widespread assessment of their atmospheric transport, lifetime, and chemical aging processes. Specifically, ATom
consisted of over 48 science flights with vertical profile sampling from 0.15 to 13 km from ~85° S to ~82°N (Thompson et al.,
2022). The flight route of this extensive mission is presented in **Fig. 1**. To capture the seasonal variability of the atmosphere,
these flight routes were performed once during each of the four seasons. The chemical composition of individual aerosols was
measured using PALMS, which has been previously deployed for aircraft-based single-particle measurements (Thomson et
al., 2000; Hudson et al., 2004; Liao et al., 2015). This mission is especially valuable for understanding the prevalence of ship
exhaust particles because most of the measurements were made over the open ocean (**Fig. 1**).
In brief, PALMS is a single-particle mass spectrometer that uses a laser desorption ionization technique to characterize both
non-refractory and refractory components of individual particles with diameters of 0.12 to 5 µm (Murphy et al., 2006). First,
individual particles are introduced into the instrument by an aerosol-focusing inlet. Second, the single particles pass through
two continuous beam lasers ($\lambda_{max} = 503$ nm) and the resultant scattered light from both continuous lasers is used to determine
the transit time of the particle, which can be used to calculate an aerodynamic diameter based on laboratory calibrations of
size-selected polystyrene latex sphere standards (Duke Scientific). Third, the scattered light from the second continuous beam
triggers a pulse from a 193 nm excimer laser, which ablates the particle and produces ions. Fourth, the generated ions pass
through a Time-of-Flight (ToF) mass spectrometer and are detected by their mass to charge ratio (m/z). During ATom, the
polarity of the mass spectrometer was switched every few minutes to alternately detect negative and positive ions. In this work,
we only use positive mass spectra, as vanadium is detected in this mode.



To classify the particles measured during ATom as PM from ship exhaust, we use spectral signatures of vanadium, nickel, iron, sulfate, and organics, which have been previously identified in single-particle mass spectrometry measurements in ship plumes near Rostock, Germany (Passig et al., 2021) and in the port of Los Angeles (Ault et al., 2010). In addition, we use empirical categories which are supported by clustering algorithms (Murphy et al., 2003) to categorize ship exhaust, biomass

burning, mineral dust, sea salt, organic/sulfate/nitrate, and meteoric particle types by grouping similar mass spectra (Froyd et al., 2019). These categorizations are not perfect and can occasionally lead to the misidentification of one particle type for another; for example, both ship exhaust and mineral dust particles can contain vanadium (Adachi et al., 1997). To prevent misclassifying PM from ship emissions for mineral dust, we filter out particle spectra with high aluminum and silicon signatures (m/z of 27 and 28, respectively), which represent the major aluminosilicate component of mineral dust (Falkovich

et al., 2001), relative to vanadium $V^+$ (m/z = 51), $VO^+$ (m/z = 67).

The average mass spectra of identified vanadium-containing ship exhaust particles sampled in the MBL during the ATom campaign are presented in Fig. 2a. Additional average mass spectra for this particle type as a function altitude are shown in Fig. S1. As shown in Fig. 2a, the average spectra appear similar across each ATom campaign, which suggests the composition of ship exhaust particles does not vary with season. For example, this particle type contains distinct spectral peaks, including

$V^+$ (m/z = 51), $VO^+$ (m/z = 67), $Ni^+$ (m/z = 58), $Fe^+$ (m/z = 56), $Na^+$ (m/z = 23), $S^+$ (m/z = 32), $SO^+$ (m/z = 48), which is reflective of the chemical composition of heavy fuel oil (Ali and Abbas, 2006; Uhler et al., 2016). In addition, the mass spectra of our identified ship exhaust particles are in good agreement with those previously reported for single-particle mass spectrometry measurements taken in fresh ship plumes (Passig et al., 2021; Ault et al., 2010). However, in contrast to these studies, we observe a strong $NO^+$ (m/z = 30) signature, which may be a result of PALMS having higher sensitivity for this

signal and/or indicative of atmospheric aging processes.

## 2.2 Laboratory calibrations for vanadium

To quantify vanadium in ship exhaust particles, we performed laboratory calibrations by nebulizing mixtures with known quantities of vanadium, sulfate, and organic compounds. We use the resultant aerosol as a proxy for PM emitted by ships and measure characteristic mass spectral signatures with PALMS. The calibration standards were prepared by dissolving varying

amounts of vanadium(IV) oxide sulfate hydrate (≥99.99%; Sigma-Aldrich) and ammonium sulfate (≥99.0%; Sigma-Aldrich) in deionized water (18 MΩ·cm) from a Millipore UV ultrapure water system. In cases where the organic content was varied in the calibration standards, a 1:1 mixture of sucrose (≥99.5; Sigma-Aldrich) and adipic acid (≥99.0%; Gold Label, Aldrich) was used. These organic mixtures have been previously used in organic single-particle mass fraction calibrations for PALMS (Froyd et al. 2019). The solutions were aerosolized using laboratory-generated zero-air gas, which was fed into an Aeromist

nebulizer.

As illustrated in **Fig. 2b**, the average mass spectra of the 0.1 wt % vanadium standard are representative of ambient ship exhaust particles measured during ATom, with the exception of Fe, Ni, K, and Na. To produce vanadium calibration curves, vanadium ion signals ($V^+$ and $VO^+$) were normalized to the sum of vanadium, organic ($C^+$, $CH^+$, $CO^+$, and $C_3^+$), and sulfate



($S^+$, $SO^+$, and $SO_2^+$) ion signals from a suite of prepared standards. We include the organic and sulfate signals in the

normalization because these species make up the bulk composition (>95%) of primary PM emitted from heavy fuel oil

combustion (Wu et al., 2018). The normalized vanadium signals for each calibration standard are then plotted as a function of

known vanadium single-particle mass fraction in the standards to obtain a calibration curve (see **Fig. S2**). This range of

vanadium content in the calibration curve (0.005 to 1.5 wt % in solution - 0.01-2 wt % normalized to sulfate and organics) is

reflective of > 85% of the vanadium ion signals observed in ship exhaust particles from ambient measurements obtained during

ATom (**Fig. 2b**).

To determine how efficiently vanadium is ionized within single particles, we obtain the relative ionization efficiency (RIE) of

vanadium to organic and sulfate by fitting the calibration data with Eq. (1) equation presented by Froyd et al., 2019:

$$mf_v = \frac{m_v}{m_v + m_{org} + m_{sulf}} = \frac{sf_v}{\beta + sf_v(1-\beta)} \qquad (1)$$

Here, $mf_v$ represents the single-particle mass fraction of vanadium, $m_v$ is the mass of vanadium in a single particle, $m_{org}$ is the

mass of organics in a single particle, $m_{sulf}$ is the mass of sulfate in a single particle, $sf_v$ is the ion signal fraction of vanadium

in a single particle, and $\beta$ is the RIE of vanadium to organic and sulfate signals. In this work, we explore the RIE of vanadium

for two conditions: 1) mixed vanadium(IV) oxide sulfate hydrate and ammonium sulfate solutions and 2) mixed vanadium(IV)

oxide sulfate hydrate and ammonium sulfate with 20 wt % organics using a 1:1 sucrose to adipic acid mixture. These calibration

curves were used to capture the range of vanadium, organic, and sulfate signals observed for ambient data (see **Fig. S2**).

As shown in **Fig. S1**, we obtain an RIE value for vanadium of ~340 for mixed vanadium(IV) oxide sulfate hydrate and

ammonium sulfate and an RIE value for vanadium of ~126 for mixed vanadium(IV) oxide sulfate hydrate and ammonium

sulfate with 20 wt % organics, which suggests the presence of organics can influence the ionization efficiency of vanadium in

single particles. Conversely, vanadium can also alter the RIE of organics relative to sulfate (in comparison to pure

organosulfate particles). Although these RIE variabilities broaden the uncertainty of the vanadium single-particle mass

fractions, we apply an RIE of 330, which lies closer to the value obtained for pure vanadium(IV) oxide sulfate hydrate and/or

ammonium sulfate mixed particles without added organics, as both our ambient data as well as previous aerosol mass

spectrometry studies have shown that accumulation mode ship exhaust particles are predominantly composed of sulfate

(Murphy et al., 2009).

**2.3 The application of laboratory calibrations to ATom data**

The laboratory calibrations described above were applied to ATom data containing spectral signatures for ship exhaust particles

(see **Section 2.1**) from the ATom data sets obtained for all four campaigns. In addition, only single-particle mass spectrometry

measurements taken out-of-cloud were used in the calibrated data, as cloud droplets can produce particle artifacts (Murphy et

al., 2004). Lastly, because land-based industrial processes (e.g., coal combustion and petroleum production) can produce

vanadium-containing particles with similar mass spectra (Schlesinger et al., 2017), single-particle data obtained during ATom



were filtered for measurements taken over water. We also exclude data poleward of 65° N and 65° S because those regions are often ice-covered and not currently relevant for shipping.

The reported number of observed ship exhaust particles in the atmosphere depends on the criteria used to define this particle type (see **Section 2.1**). Using more stringent criteria will reduce the total number of ship exhaust particles observed in the

atmosphere during ATom; however, this will also lead to fewer misclassifications of mass spectra that may marginally resemble ship exhaust PM. In this work, we use strict criteria for the classification of ship exhaust particles, in which vanadium peak signals (m/z = 51 and 67) must be at least three times greater than the sum of the surrounding peak signals, as well as five and two times greater than aluminum and silicon peaks, respectively to remove potential mineral dust particles. We note that the application of this particle criteria results in a more conservative estimate for the number of ship exhaust particles

measured during ATom.

Despite this relatively strict definition for ship exhaust particles, we do not have confidence in the identification of ship exhaust particles during very high mineral dust loadings because some mineral dust particles contain vanadium with low aluminum and silicon ion signals. Therefore, we exclude measurements made in the Saharan Air Layer (SAL) dust region over the Atlantic, which ranges from -5° S to 30° N (-5-30° latitude) and 10° to 30° W (-10-30° longitude) depending on the season

(Tsamalis et al., 2013).

The largest factor that contributes to an underestimation of the reported number of ship exhaust particles in this study is the inability of PALMS to measure particles < 120 nm, which make up a significant component of ship exhaust particles (Zhou et al., 2019; Murphy et al., 2009). For example, Zhou and colleagues investigated the particle size distributions (10 nm to 10 µm) of exhaust particles from various marine engines with different fuel types and observed that the majority of particles were <

100 nm in size (Zhou et al., 2019). These ultrafine particles still make up a significant fraction of the total number of particles in plumes that are over an hour old (Murphy et al., 2009), which suggests that these particles likely exist long after a ship has passed.

## 3 Results and discussion

### 3.1 Ship exhaust particles are not limited to the marine boundary layer

As shown in **Fig. 3**, ~1% of accumulation mode particles ($d_p$ = 0.12-1 µm) measured by PALMS at altitudes < 2 km are particles emitted by ships. This percentage decreases with increasing altitude, which suggests that PM from ships is removed via wet deposition processes. The removal of these particles via wet deposition is consistent with previous work by Coggon et al., 2012, who reported enhancements in both vanadium content in stratocumulus cloud water samples and cloud droplet number in marine air influenced by emissions from cargo and tanker ships. The processes and properties behind clouds formed

by these ship exhaust particles (i.e., ship tracks) have been well documented in climate studies over the last several decades due to their indirect cooling effects from light scattering, which is suggested to reduce Earth's radiative forcing by 0.02-0.27 W m$^{-2}$ (Yuan et al., 2022).



Although ship exhaust particles play an important role in the formation of clouds, not all these particles are scavenged by water droplets during atmospheric convection. For example, vanadium-containing ship exhaust PM still makes up ~0.1% of the accumulation mode particles above 12 km (**Fig. 3**), which is approximately an order of magnitude lower than the number fractions measured in the marine boundary layer (MBL). Within the MBL, the highest number fraction of ship exhaust particles is in the Northern Hemisphere of the remote central Pacific and Atlantic Ocean between ~0 and ~50 degrees latitude (**Fig. 4a**), which is reflective of major shipping routes (Wu et al., 2017). At altitudes above the MBL, the number fraction of vanadium-containing ship exhaust particles decreases for both hemispheres (**Fig. 4b**). These results suggest that vanadium-containing ship exhaust particles are sufficiently widespread in the atmosphere

It is important to mention that although ship exhaust plumes were not targeted during ATom, we encountered several sampling episodes of air heavily populated with vanadium-containing particles that were correlated with elevated numbers of black carbon, accumulation mode, and nucleation mode particles. These air masses also contained higher than background levels of $NO_y$ and $SO_2$ mixing ratios, which suggests that we were likely sampling ship plumes. The encountered ship plumes appear to be somewhat dilute as they also contain particles from marine origin; for example, sea salt particles can contribute up to 60% of total measured particles during these sampling periods. These dilute plume events can lead to unexpectedly noisy data when determining the widespread distribution of ship exhaust PM as one plume sampling episode can lead to an increase in the total fraction of vanadium-containing particles for a given area. For example, although the number fraction of vanadium-containing ship exhaust particles generally becomes more uniform above the MBL for a given hemisphere, these diluted plume events can lead to elevated number fractions within and above the MBL. This is particularly true for ATom 1, where the highest number of vanadium-containing particles in and above the MBL are observed at ~30° N in both the Atlantic and Pacific Oceans (**Fig. 4**). This is a result of the aircraft sampling the same vanadium-enriched air masses at different altitudes during vertical profiling.

The diluted plume sampling events allow us to infer the fraction of ship exhaust particles that contain vanadium. During these episodes, organic/sulfate, sea salt, vanadium, and soot particles were all detected using PALMS. Aside from sea salt, these particle types have been previously identified in ship plumes (Ault et al., 2010). Using particle type fractions obtained from PALMS data during dilute plume measurements, we estimate that vanadium-containing particles can contribute anywhere from 10-40% of ship exhaust particles in the accumulation mode. This is in good agreement with Passig et al., 2021, who reported that 10-20% of particles during transient ship plume events off the coast of Rostock, Germany contained vanadium.

**3.2 Vanadium content of individual ship exhaust particles**

As shown in **Fig. S4**, ship exhaust particles can have over an order of magnitude difference in vanadium content with most of the particles containing < 2 wt % vanadium. In addition, the highest vanadium single-particle mass fractions are observed in the MBL compared to the free troposphere (**Fig. S3**). Since vanadium is stable in the particle phase, we do not expect the absolute amount of vanadium to change as a function of altitude. Instead, these results indicate that ship exhaust particles experience atmospheric aging through the accumulation of non-vanadium species, which would reduce the contribution of the



vanadium ion signals to the total ion signals obtained for individuals and lead to lower vanadium single-particle mass fractions. Further discussion on the atmospheric processing of ship exhaust particles is discussed in **Section 3.4**.

The vanadium single-particle mass fractions in the Pacific and Atlantic Oceans in the northern hemisphere 20° N to 65° N (20 to 65° latitude), tropics 20° S to 20° N (-20 to 20° latitude), and southern hemisphere 20° S to 65° S (-20 to -65° latitude) are

presented in **Fig. 5.** The vanadium single-particle mass fractions in this figure are obtained from the average vanadium content of all the ship exhaust particles sampled in a given area (i.e., northern hemisphere, tropics, and southern hemisphere). We measure the highest average vanadium single-particle mass fraction for particles sampled in the tropical Pacific during summer compared to other locations and seasons. The majority of particles measured in this tropical Pacific region were sampled from a single plume event that contained higher than average vanadium ion signals measured throughout ATom. These vanadium

single-particle mass fractions are also higher than those measured in other ship plume events, which may be attributed to this particular ship using a fuel that contained more heavy metal impurities or having different engine operating conditions.

### 3.3 Ship exhaust particles are rich in sulfate and lower in organic matter, ammonium and nitrate

To determine the organic and sulfate single-particle mass fractions in individual ship exhaust particles, we apply the organic and sulfate calibrations discussed in **Section 2.2**. As illustrated in **Fig. 6**, ship exhaust particles contain more than 80% sulfate.

These results agree with previous studies investigating the chemical composition of ship exhaust particles (Ault et al., 2010; Murphy et al., 2009; Healy et al., 2009; Agrawal et al., 2008); for example, using aircraft-based aerosol mass spectrometer measurements on the Twin Otter, Murphy and colleagues determined that ship exhaust particles contain over 70 wt % sulfate (Murphy et al., 2009). The dominant sulfate fraction of ship exhaust particles determined from in-situ measurements is compositionally different from the heavy fuel oils used in marine vessels, which are predominantly composed of organic

material and only up to a few wt % sulfur (Uhler et al., 2016). This chemical difference has been attributed to the aqueous oxidation of $SO_2$, a major gas-phase pollutant emitted during ship fuel combustion, to particle-phase sulfate and sulfuric acid (Healy et al., 2009; Murphy et al., 2009). The aqueous-phase oxidation of $SO_2$ in the atmosphere is suggested to happen quickly (Finlayson-Pitts and Pitts Jr., 1999), which explains why ship exhaust particles measured in fresh plumes are still sulfate-rich (Murphy et al., 2009). Finally, previous studies have demonstrated that vanadium can participate in the catalytic oxidation of

$SO_2$ (Barbaray et al., 1978). Similar catalytic oxidation mechanisms may also be true for other transition metals in ship exhaust particles, including iron (Fu et al., 2007).

Although sulfate is the dominant component of the vanadium-containing ship exhaust particles measured in this study, there is still < 20 wt % organic component. Similar to previous aerosol and single-particle mass spectrometer measurements (Liu et al., 2022), the organic fraction of ship exhaust particles appears to be relatively oxidized throughout ATom. For example,

organic aging indicators, including oxalate ($C_2O_4^+$) and other oxidized organic signals (e.g., $CO^+$, $C_2H_3O^+$) do not vary with altitude. Furthermore, these signals appear similar for in and out of dilute plume measurements, which suggests the majority of organic oxidation occurs relatively quickly. These results agree with the single-particle aerosol mass spectrometer measurements made offshore of the East China Sea, which demonstrated high degrees of oxidation for vanadium-containing



ship exhaust particles (Liu et al., 2022). We also observe oxidized organic ion signals in non-vanadium-containing ship exhaust

particles; for example, both organic/sulfate and soot particles measured in dilute plume events during ATom often displayed

$CO^+$ ion signals in their mass spectra.

As illustrated in **Fig. 2**, ammonium ion signals ($NH_4^+$) were consistently detected in ship exhaust in single-particle mass spectra

of ship exhaust particles during ATom. Ammonium has been previously reported in ship exhaust particles and is a secondary

species that is formed from the reaction of ammonia with sulfuric acid that is formed during $SO_2$ oxidation (Ault et al., 2010).

The PALMS instrument detects neutralized and incompletely neutralized sulfuric acid in positive ions as $SO^+$ and $H_sSO_4H^+$.

During ATom, we detect both $SO^+$ and $H_sSO_4H^+$ in vanadium-containing particles, which is consistent with incomplete

neutralization of a substantial sulfate/sulfuric acid content with ammonia. Using the $NH_4^+$ ion signal from laboratory

calibrations, we estimate that vanadium-containing ship exhaust particles measured during ATom contain less than 16 wt %

of ammonium.

Although $NO^+$ signals can be produced from ammonium during laser ablation, it is also formed from nitrate; therefore,

ammonium may not be entirely responsible for the $NO^+$ ion signal observed in ambient spectra. In fact, nitrate has been

previously measured in ship exhaust particles near ports (Healy et al., 2009) and in ship plumes (Murphy et al., 2009) and is

thought to be formed by secondary reaction mechanisms. The formation of particulate-phase nitrate in aged ship exhaust

particles has been explained by ozone oxidation of $NO_x$ from a field study conducted at a port in Shanghai China (Wang et al.,

2019). In this work, the authors used wind diagrams to determine fresh and aged particles from ship plumes and found that

aged particles contained higher nitrate ion signals in regions of depleted ozone mixing ratios. Although ozone measurements

were made during ATom, it is difficult to correlate the nitrate content of ship exhaust particles with ozone mixing ratios, as

ozone becomes depleted in the MBL due to ocean surface deposition and entrainment in sea spray aerosol (Singh et al., 1996).

Nitrate can also form on ship exhaust particles via the direct reduction of $NO_x$ through catalytic heterogeneous chemistry with

vanadium and other multi-valent metals (e.g., iron, nickel), which will be discussed in more detail in the next section (**Section

3.4**).

## 3.4 Metal catalysis in ship exhaust particles

Many of the metals generated during the combustion of heavy fuel oil are multivalent (e.g., vanadium, nickel, and iron) (Corbin

et al., 2018), which means they have the potential to participate in redox reactions with other atmospheric pollutants. For

example, nitrate formation has been previously observed during $NO_x$ uptake by mineral dust and mineral dust proxies (i.e.,

metal oxides) (Underwood et al., 1999). The metals in ship exhaust particles may similarly undergo redox chemistry with $NO_x$

to lead to the production of particulate nitrate. Furthermore, if vanadium, iron, and nickel are present in their semiconducting

metal oxide forms (i.e., $V_2O_5$, NiO, $Fe_2O_3$) then their optical band gaps would lie within the UV/visible region of the solar

spectrum (Beke, 2011; Hosny, 2011; Al-Gaashani et al., 2013), which could lead to photoenhanced decomposition of $NO_2$ and

the subsequent formation of nitrate. This photochemical transformation has been previously observed for mineral dust and



metal oxides (Ndour et al., 2009, 2008; Chen et al., 2011); however, we note that the chemistry of the metal species in ship exhaust particles may be different than that of mineral dust, as their chemical form and solubility is not well understood. Similarly, particulate sulfate can be formed via the heterogeneous chemistry of transition metals with $SO_2$. For example, work by Barbaray and coworkers demonstrated that vanadium pentoxide particles can catalyze the oxidation of $SO_2$ into particle-phase sulfate in the presence of $H_2O$ and $NO_2$ (Barbaray et al., 1978). This catalytic oxidation mechanism has been previously hypothesized to occur in PM emitted by ships (Murphy et al., 2009; Ault et al., 2010) and was further supported by a study that determined vanadium pentoxide to be the dominant vanadium species of fine PM in diesel exhaust (Shafer et al., 2012). Similarly, iron and manganese have been found to catalyze the oxidation of $SO_2$ (Hoffmann and Jacob, 1984) and are suggested to be responsible for 9-17% of global sulfate production (Alexander et al., 2009).

Transition metal ion catalyzed reactions with $H_2O_2$ in aqueous particles can lead to the formation of reactive oxygen species (ROS), including the hydroxyl radical (OH) and peroxy radical ($HO_2$) (Lousada et al., 2012; Enami et al., 2014). This Fenton or Fenton-like chemistry is suggested to play an important role in $H_2O_2$ atmospheric budget and can influence the oxidative capacity of the atmosphere. Iron is the most important transition metal for controlling the OH, $HO_2$, $H_2O_2$ budget and is thought to be the main driver of $H_2O_2$ uptake in ambient aerosols (Qin et al., 2022). This ROS budget is thought to be controlled by the number concentration of iron-containing particles rather than the total iron mass fraction (Khaled et al., 2022), which highlights the importance of single-particle chemical analysis for determining the oxidative capacity of the atmosphere. Although this study focused on iron-catalyzed chemistry with $H_2O_2$, vanadium-containing ship exhaust particles almost always contain iron (Healy et al., 2009) and could provide clues to the importance of Fenton chemistry for ship exhaust particles. Finally, other transition metals, including copper (Song et al., 2020) and vanadium (Mizuno and Kamata, 2011) have also been found to react with $H_2O_2$ to produce ROS. Given the prevalence of metals in ship exhaust particles, it is reasonable to assume that these particles can also facilitate the formation of ROS species through interaction with $H_2O_2$.

**3.5 Vanadium-containing ship exhaust particles experience atmospheric aging**

As described above, the vanadium single-particle mass fractions calculated in this work were normalized to the sum of the vanadium, sulfate, and organic signals (see **Section 2.2**); therefore, a decrease in vanadium single-particle mass fraction must be correlated with an increase in the organic and/or sulfate ion signals. As shown in **Fig. 6**, we observe an increase in the sulfate single-particle mass fraction in ship exhaust particles with increasing altitude, which suggests ship exhaust particles continue to accumulate sulfate during their atmospheric lifetime. These results are in good agreement with those presented by Murphy and coworkers, who observed a gradual increase in sulfate mass fractions of ship exhaust particles as a function of plume age using an aerosol mass spectrometer (Murphy et al., 2009). However, we do not expect this few wt % increase in sulfur single-particle mass fraction observed during atmospheric aging to significantly impact the cloud droplet activation properties of these particles as they are predominantly composed of sulfate immediately after emission (Murphy et al., 2009). As illustrated in **Fig. 7**, the $NO^+$ signal measured for individual ship exhaust particles increases as a function of altitude. Previous studies have associated nitrate in ship exhaust particles with more aged particles (Wang et al., 2019; Passig et al.,



2021). Although we did not attempt to quantify nitrate single-particle mass fractions in individual ship exhaust particles in this
work, Murphy and coworkers determined that nitrate in the non-refractory dry component of ship exhaust particles increased
up to 1 wt % in plumes that were over an hour old (Murphy et al., 2009).

## 4 Conclusion

In this study, we report that ~1 % of particles measured in the MBL over the remote Atlantic and Pacific are vanadium-
containing particles from ship emissions from measurements that did not target ship plumes. In addition, we use laboratory
calibrations to determine that the majority of these particles contain less than 2 wt % vanadium but that the single-particle
mass fractions in individual particles can vary over an order of magnitude. Furthermore, we demonstrate that these particles
are predominantly made up of sulfate (> 80 wt %) and continue to accumulate sulfate, as well as nitrate during atmospheric
aging. Finally, we note that there are likely more vanadium-containing particles that are smaller than the lower size limit of
our measurements, which may lead us to underestimate the fraction of particles in the marine atmosphere.
Although vanadium-containing ship exhaust particles are primarily composed of sulfate, the few wt % or less of vanadium in
these particles can influence catalytic heterogeneous chemistry and photochemistry of ship exhaust particles, which can change
their chemical composition and cloud formation properties. For example, vanadium-based catalysts have been explored as a
commercial material for denitrification of $NO_x$ in combustion plumes (Chen et al., 2018). These catalysts have been shown to
be effective in removing $NO_x$ with less than 1 wt % of vanadium. Despite the powerful catalytic capabilities of vanadium and
other transition metals, it is difficult to determine how much of the nitrate and sulfate accumulation in ship exhaust particles
is attributed to redox chemistry with these metallic species versus other aqueous phase oxidation reaction mechanisms.
Finally, we note that the measurements presented in this work were conducted before the International Maritime
Organization's Sulfur 2020 policy, which required the shipping industry to transition from 3.5% to 0.5% m/m sulfur-
containing fuel. This policy, which has demonstrated effectiveness in decreasing $SO_2$ emissions from the Maritime Industry
(Kim et al., 2022; Song et al., 2022), may also alter the current distribution and single-particle mass fractions of vanadium in
ship exhaust particles. For example, Yu and colleagues reported significant reductions in vanadium loadings from bulk ship
PM measurements conducted after Sulfur 2020 (Yu et al., 2021); however, vanadium is still frequently detected in ship
exhaust particles emitted during the combustion of low sulfur fuels (Zhou et al., 2020). It is currently not clear how low
sulfur fuels impact the vanadium content on an individual particle level and whether the single-particle mass fractions are
still sufficiently large for ship particles to participate in catalytic heterogeneous chemistry and cloud droplet activation.



a)

b)

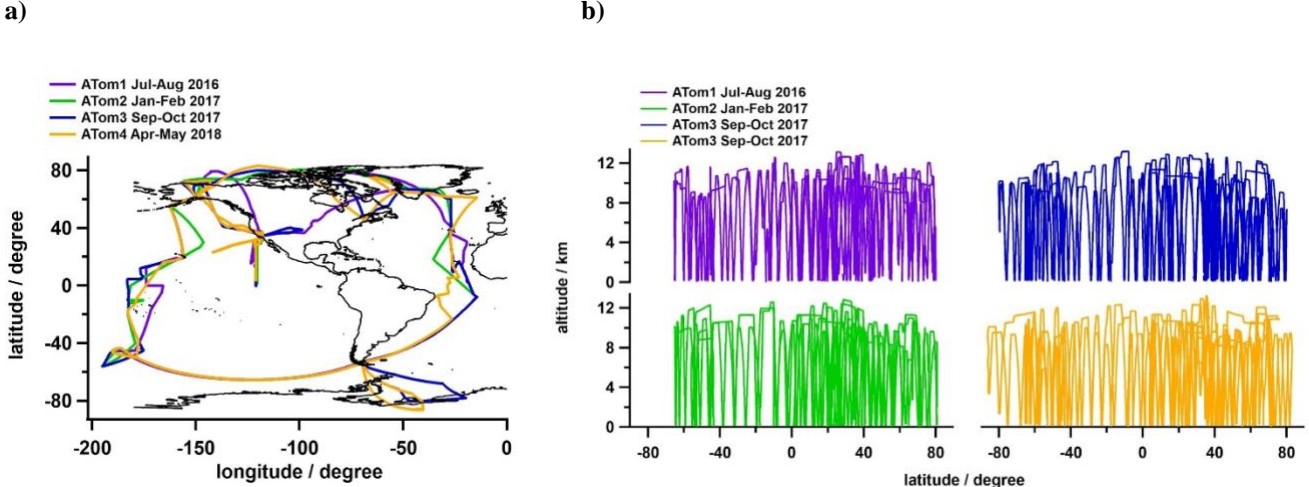

**Figure 1: a) The flight route of the NASA DC-8 aircraft during each of the four ATom campaigns and b) sampling altitudes as a function of latitude during ATom. Each colored line represents the flight path and altitude taken for a given campaign.**






a)    b)

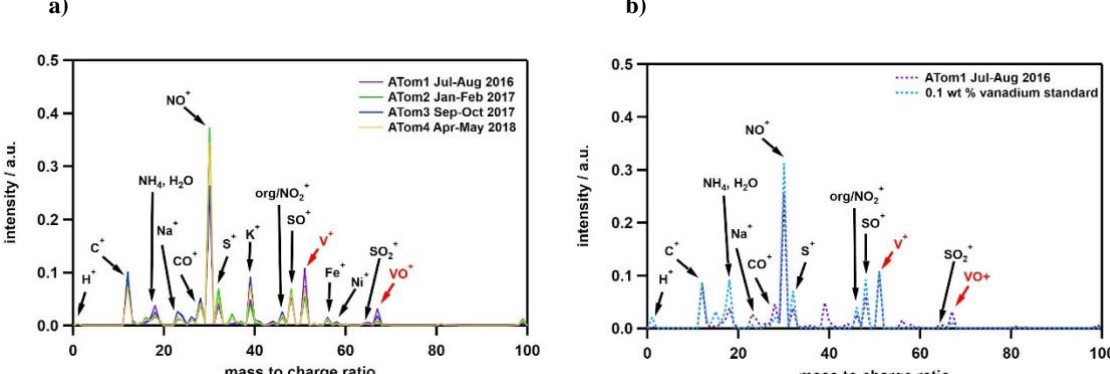

**Figure 2: Mass spectra degraded to unit mass resolution for average mass spectra comparisons of vanadium-containing ship exhaust particles measured in the MBL a) during all ATom campaigns b) during ATom 1 with a 0.1 wt % vanadium calibration standard. Only similar peaks between ATom and the calibration standard are labeled. The standard was a mixture containing vanadium sulfate, ammonium sulfate, and trace organics. The V+ peak for ATom 1 is difficult to see due to overlap with the calibration standard.**




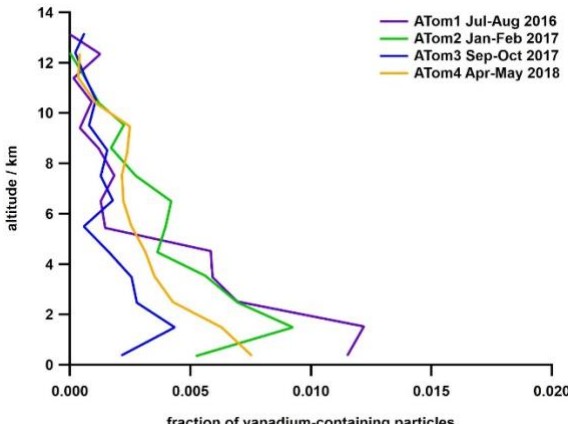

**Figure 3: Fraction of vanadium-containing particles as a function of altitude. Each colored line represents a different ATom campaign with 20 particle averages.**






a)                                          b)

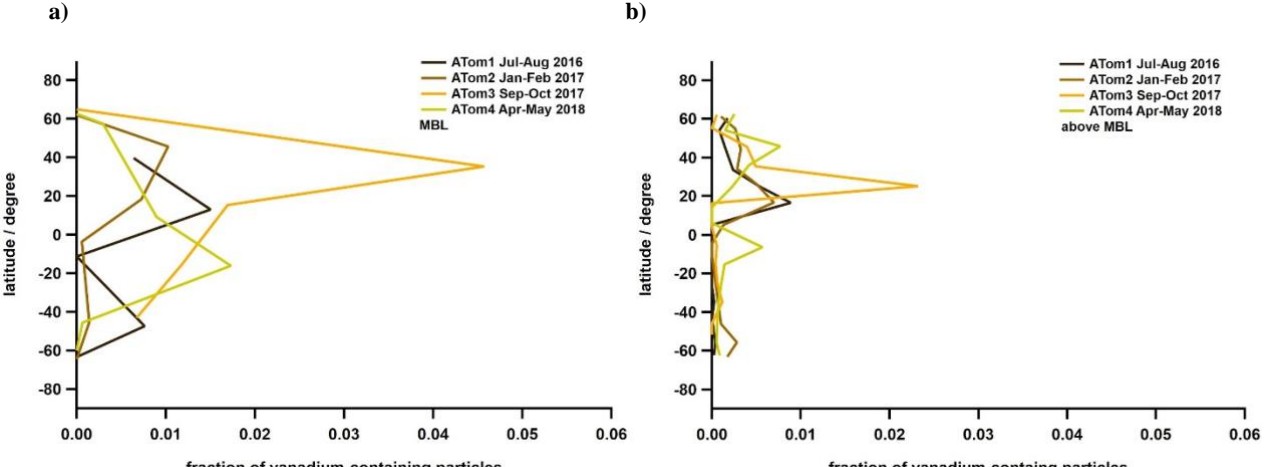

**Figure 4: Fraction of vanadium-containing particles as a function of latitude in a) the MBL and b) above the MBL. Each colored line represents a different ATom campaign and oceanic region with 20 particle averages. We note that the fraction of vanadium-containing particles presented below exclude measurements taken in the Saharan Air Layer off the west coast of Africa to filter out vanadium-containing mineral dust particles. In addition, we exclude measurements made above 65° N and below 65° S to filter polar regions, which lack ship traffic.**






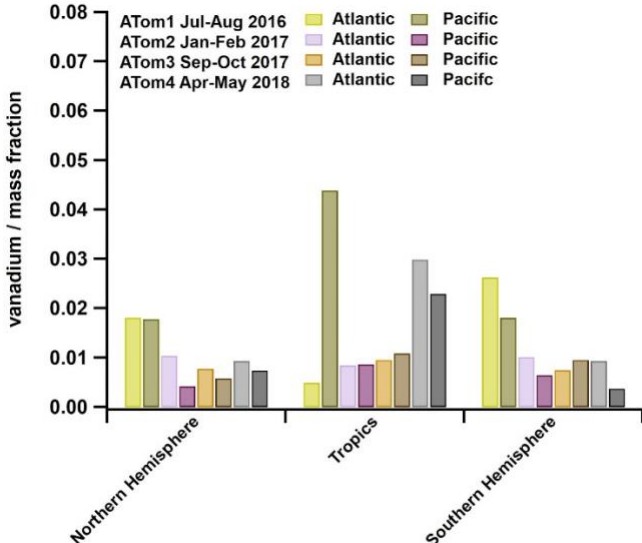

**Figure 5: Average vanadium mass fractions for vanadium-containing ship exhaust particles measured in the Northern Hemisphere (20° N to 65° N; 20 to 65° latitude), Tropics (20° S to 20° N; -20 to 20° latitude), and Southern Hemisphere (20° S to 65° S; -20 to -65° latitude) in the Atlantic and Pacific Ocean for all four ATom at all sampling altitudes.**



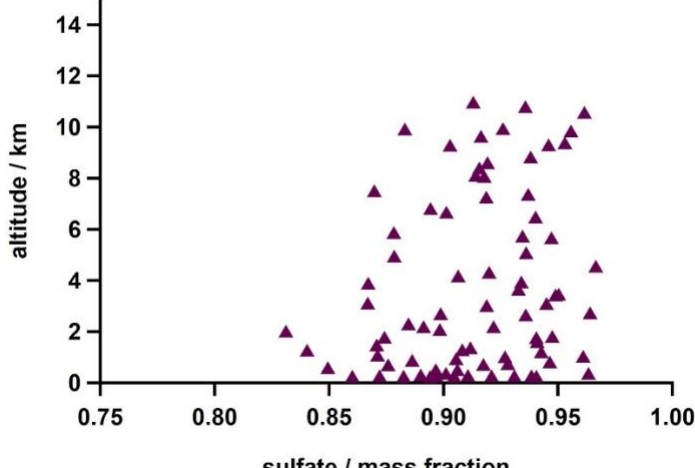

**Figure 6: Sulfate mass fractions in vanadium-containing ship exhaust particles as a function of altitude for ATom 1-4. Each point represents 20 particle averages.**





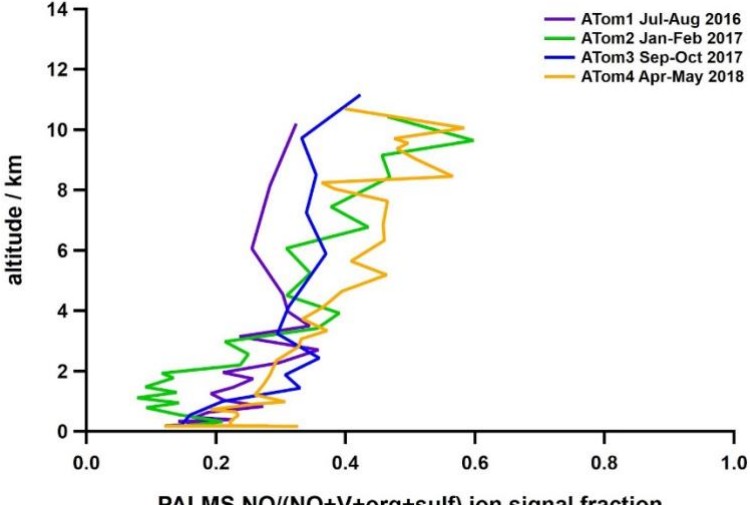

**Figure 7: NO⁺ ion signals as a function of altitude for vanadium-containing ship exhaust particles measured during ATom. Each colored line represents a different ATom campaign with 20 particle averages.**





**Data Availability**

Data are publicly available at https://daac.ornl.gov/ATOM/guides/ATom_merge.html.

**Supplement**

**Author contributions**

M.A-G. conducted laboratory aerosol calibrations with assistance from M.J.L. M.A-G. performed data analysis and wrote the manuscript with critical comments from D.M.M, G.P.S, M.J.L., and K. D. F. PALMS data was collected during research flights by D.M.M., G.P.S, and K.D.F.

**Competing interests**

The authors declare no competing interests.

**Acknowledgements**

M.A-G. held an NRC Research Associateship award at NOAA Chemical Sciences Laboratory.

**Financial support**

The Atmospheric Tomography Missions were supported by NASA's Earth System Science Pathfinder Program EVS-2 funding. Research flight participation for D.M.M., G.P.S, and K.D.F was supported by NOAA climate funding and NASA (no. NNH15AB12I). NOAA cooperative agreement NA17OAR4320101 supported G.P.S., K.D.F., and M.J.L. NOAA cooperative agreement NA22OAR4320151 supported G.P.S. and M.J.L. An NRC Research Associateship award at NOAA Chemical Sciences Laboratory was held by M.A-G.




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
