# Peer review of "Measurement report: Vanadium-containing ship exhaust particles detected in and above the marine boundary layer in the remote atmosphere"

_EGUsphere, 2023_

## Author Comment (AC1)

**Measurement report: Vanadium-containing ship exhaust particles detected in and above the marine boundary layer in the remote atmosphere–reviewer comments**

Maya Abou-Ghanem[1], Daniel M. Murphy[1], Gregory P. Schill[1], Michael J. Lawler[1,2,] and Karl D. Froyd[1,2]

[1]Chemical Sciences Laboratory, National Oceanic and Atmospheric Administration, Boulder, 80305, USA
[2]Cooperative Institute for Research in Environmental Sciences, University of Colorado Boulder, Boulder, 80309, USA

We would like to thank the reviewers for their insightful comments and suggestions, which have enhanced the quality and clarity of the paper.

In addition, changes requested by the reviewers, we have made the definition of the Saharan Air Layer more consistent between figures in the manuscript. The changes are minor and do not affect the discussion or conclusions.

**REVIEWER 1**

**General comments:**

The paper entitled "Vanadium-containing ship exhaust particles detected in and above the marine boundary layer in the remote atmosphere" analyzed the PALMS instrument of the NASA DC-8 aircraft during 2016-2018. The authors found that PM emitted by ships is widespread in the atmosphere, and also demonstrated that vanadium-containing ship exhaust particles were observed up to 13 km because not all ship exhaust particles are removed by wet deposition. Overall, the manuscript is well-written and concluded an important result for the air quality/climate, and health impact. I have several specific points as follows, and please address these questions and comments.

**Specific comments:**

- Line 9 (Abstract): Here the authors introduced 1.2 Tg ship emissions, whereas it is introduced as 1.67 Tg in Line 31. This might indicate different contents (I am not sure about PM10 or PM2.5 in the abstract), but it will be better to be consistent within this manuscript.

We have revised the abstract (line 9) to 1.67 Tg, which is consistent with the text in the introduction.

- Line 21-23 (Abstract): If possible, the short comment on the IMO 2020 sulfur regulation will be important attention to the readers.

We have mentioned the IMO 2020 sulfur regulation in final sentence of the abstract (line 23) as follows:

We note that this data was collected prior to the 2020 International Maritime Organization (IMO) sulfur regulation but stands as a valuable reference for understanding how ship emissions have evolved in light of these regulations.

- Line 55-57: Needs appropriate reference(s) regarding this analysis to support the importance of ship emissions.

This sentence about ship emissions refers to the data described later in the paper rather than a conclusion from the literature. We have added the Thompson et al., 2022 reference for the ATom mission on line 57.

- Line 96-105: In Line 102 to 105, how to distinguish the mineral dust has been presented. However, I understand that the authors used V, Ni, Fe, SO4, and organics for ship exhaust. For example, SO4 may also produced via anthropogenic sources and volcanoes. How to detect the ship emissions? More information is required here.

In this section, we describe the exclusion of particle spectra with high aluminum and silicon signatures, which are reflective of crustal material. This would also include the exclusion of volcanic dust in our analysis. We have also added the following on line 108 to help clarify what is meant by ship sulfate and organics:

"Note that when we refer to sulfate and organics on ship exhaust particles, that is different than the total amount of particulate sulfate and organics from ships. Sulfate and organics from ship emissions also condense onto other particles and, as discussed in section 3.5, sulfate and organics from other sources can be added to ship exhaust particles during atmospheric aging."

- Line 173-175: I am also wondering about the effect of Asian dust when the DC-8 flight passed over the Pacific Ocean. Was there no dust event in Asia during this analyzed period?

Although we did measure mineral dust over the Pacific Ocean, we did not observe any large dust events in the Pacific Ocean that would lead to misidentification of ship particles in our analysis. Only very large dust events, like the Saharan Air Layer, were problematic.

- Line 186-187: For heavy metals, is there no possibility of the removal by dry deposition process?

We have added "above the boundary layer" to the sentence on line 197 to clarify that we are referring to altitudes where dry deposition to the ocean surface is not important. The ship exhaust particles described in this study are predominantly composed of sulfate (>80%), which makes them very hygroscopic.

- Line 196: From here, the analysis focusing on MBL has been discussed; however, there seems to be no explicit definition of MBL. How to calculate MBL? Did the author assume a fixed altitude to consider MBL?

We have added a reference to a previous that defines the MBL during ATom by Brock et al., 2021 to the methods section on line 113 as follows:

The MBL was defined using the definitions in Brock et al. (2021).

- Line 196-200: In terms of this kind of analysis, how about calculating the ratio of the fraction of V-containing particles within MBL and above MBL (Fig. 4(a) divided by Fig. 4 (b))? This might draw an important suggestion for the spread of V-containing particles in the entire atmosphere.

We created this plot and it does not add much information that is not already in Figure 3, which shows information about the vertical spread of V-containing particles.

- Line 205: I do not follow how to estimate this 60% contribution.

The 60% contribution is from our single particle data. We have clarified this in line 217 of the text:

"The encountered ship plumes appear to be somewhat dilute as they also contain particles from marine origin; for example, our data show that sea salt particles can contribute up to 60% of total measured single particles during these sampling periods."

- Line 233-234: If this point is critical to mention the regional (Atlantic/Pacific and Northern/Southern Hemisphere) characteristics, how about including the information of available data numbers to calculate the V/mass fraction in this figure? If the analyzed data numbers are similar over all regions, the comparison will be meaningful, but such comparison could be meaningless under the different data numbers.

We have added a description of the number of vanadium-containing particles on line 210 with the discussion of Figure 4.

"Each ~12° latitude bin in Figure 4 includes about 1000 to 4000 particles in the MBL and 4000 to 10000 particles above the MBL during each deployment. With about 1% of particles containing vanadium and less than that above the MBL, each point in **Fig. 4** represents tens of vanadium-containing particles."

This should also give an idea of the number of particles in other figures.

- Line 236: It is also noticed that the mass fraction of vanadium was almost the same level over the northern and southern hemispheres. In general, the northern hemisphere could be polluted rather than the southern hemisphere. Did this result suggest the ship exhaust will have an impact on entire the globe? This point is not discussed, but it will be helpful for a detailed discussion.

The reviewer may have misunderstood the mass fractions. We have clarified in the caption to Figure 5 to include that the mass fractions are just for vanadium-containing particles. The Southern Hemisphere is indeed less polluted; it contains fewer particles with vanadium but the particles that do have vanadium have a similar amount everywhere. Near line 113 we state that:

"Additional average mass spectra for this particle type as a function altitude are shown in **Fig. S1**. As shown in **Fig. 2a**, the average spectra appear similar across each ATom campaign, which suggests the composition of ship exhaust particles does not vary with season."

- Line 241-243: I agree that high sulfur content is consistent with the previous study of Myrphy et al., but the data in Figure 6 pointed out almost all (except three data) of data in this study included 85% sulfate. This value seems to be higher than previous study. Is there some discussion (such as regional difference and/or sampling period) regarding this difference?

Murphy et al. 2009 demonstrated that the sulfate content increases with increasing plume age, which is a result of $SO_2$ oxidation. In their study, they tracked sampling time with plume age. In our study, the age of the plumes was not known and may be older than those sampled in Murphy et al., which may explain the higher sulfur content. We have amended the following text on line 246:

The slightly higher sulfur content presented in this work may be attributed to variations in plume ages as the measurements did not specifically target recently emitted ship plumes.

- Figure 1(b): The legend for the yellow color will be a typo in their name and measurement period. Please confirm. In addition, the altitude is converted into AGL? or ASL? I guess that this will be unified in all analyses and figures, but please clarify the unit when introducing Figure 1.

Yes, there is a typo for the yellow legend. This should be "ATom4 Apr-May 2019" and has now been revised. Altitude is ASL. The data in this paper are from above the oceans, so there is little difference between ASL and AGL.

- Figure 4: For a clear reading, it may be better to unify four colors (each season) to be consistent with Fig. 3.

We have unified the colors across all figures, not just Figures 3 and 4.

- Figure 6: Why this analysis is not divided into four ATom campaigns? Were there no distinguishment by campaigns?

That is correct. All ATom campaigns appeared similar. This is now stated in the caption.

- Line 178 and Line 180: The author's name of Zhou is redundant information in this sentence.

We've removed the reference for Zhou et al. 2019 at the end of the sentence on line 190 as they are already mentioned in the previous sentence:

"For example, Zhou and colleagues investigated the particle size distributions (10 nm to 10 µm) of exhaust particles from various marine engines with different fuel types and observed that the majority of particles were < 100 nm in size."

- Line 223: "Fig. S3" seems to be "Fig. 3". Please confirm.

That is correct. We've changed the text on line 235 to **Fig. 3**.

Finally, we apologize for the slow response to reviewers. The lead author started a new job during the review process, which led to some delays in our ressponse times.

---

## Author Comment (AC2)

**Measurement report: Vanadium-containing ship exhaust particles detected in and above the marine boundary layer in the remote atmosphere–reviewer comments**

Maya Abou-Ghanem[1], Daniel M. Murphy[1], Gregory P. Schill[1], Michael J. Lawler[1,2,] and Karl D. Froyd[1,2]

[1]Chemical Sciences Laboratory, National Oceanic and Atmospheric Administration, Boulder, 80305, USA
[2]Cooperative Institute for Research in Environmental Sciences, University of Colorado Boulder, Boulder, 80309, USA

We would like to thank the reviewers for their insightful comments and suggestions, which have enhanced the quality and clarity of the paper.

In addition, changes requested by the reviewers, we have made the definition of the Saharan Air Layer more consistent between figures in the manuscript. The changes are minor and do not affect the discussion or conclusions.

**REVIEWER 2**

'**General comments:**

The authors present data on the distribution of vanadium-containing ship exhaust particles in the atmosphere, measured during regular flights over large areas of the world's oceans at different altitudes. This is a unique dataset and the authors report on the prevalence of these particles also in remote regions and high altitudes. Ageing mechanisms and the chemistry of vanadium-containing particles are discussed in the context of the altitude-dependent particle number fractions and their chemical composition.

The study provides valuable insights into the distribution of ship emission particles, prerequisite for a better understanding of their climate impact. The manuscript is technically sound and interesting. There are a few points that could help to further improve the manuscript:

**Specific comments:**

- When emphasizing the importance of ship emissions, the authors could consider to also provide some newer literature, that also addresses the current changes due to the fuel regulations, e.g. Kuittinen et al. Environ. Sci. Technol. 2021, 55, 1, 129–138, Jonson et al., Atmos. Chem. Phys., 2020, 20, 11399 —11422, Anders et al., Environ. Sci.: Atmos., 2023, 3, 1134-1144 etc.

We have added the Anders et al., 2023 reference to line 48. Although the other citations you have mentioned are insightful, they are less relevant to this manuscript. We do cite newer literature on vanadium-containing ship exhaust particles near line 71:

"These methods can also result in a reduction in the total number of vanadium-containing particles emitted by ships (Yu et al., 2021; Xiong et al., 2023); however, the reductions are not complete as vanadium-containing ship exhaust particles are still detected in ECAs (Passig et al., 2021; Xiong et al., 2023)."

- line 30: In the abstract, the -mount of annual PM emissions was given as 1.2 Tg/y, here as 1.67 Tg/y PM.10. Also, please provide a reference.

This abstract text was changed to 1.67 Tg in line 9. The following reference was added to line 30:

Additionally, the combustion of heavy fuel oil from ships contributes to approximately 1.67 Tg y-1 of particulate matter with an aerodynamic diameter of 10 µm or smaller (PM10) (Eyring et al., 2005).

- line 110: Traditionally, the particle's sulphur content is evaluated by the negative charged sulphate ions and sulphuric acid. I understand that the compact aircraft-deployable design of PALMS only allows for unipolar measurements, but could the authors provide a brief statement on the evaluation of sulphur content via S+ and SO+, particularly regarding potential interference with carbon cluster from soot at m/z=48?

Although soot can appear at m/z=48, soot spectra can be distinguished by the presence of $C_n$, $C_nH$, and $C_nH2$ (n = 1, 2, 3, etc.), which were not observed in the vanadium-containing particle spectra identified in this work. This is now stated in the manuscript on line 118:

"$SO^+$ at m/z = 48 can be distinguished from $C_4^+$ from soot by the absence of peaks such as $C_3^+$ and $C_4H^+$."

- line 113-115: The strong NO+ signal compared to e.g. Ault (2010) and Passig (2021) can be attributed to the wavelength of the used ArF-Excimer laser. In a direct comparison of a 248 nm KrF-laser with a 193 nm ArF laser, it could be shown that the 248 nm laser is much more sensitive to iron and transition metals due to a resonance effect, but the 193 nm laser is more effective in ionizing NO+ and nitrogen-containing organics, see Passig et al., ACP, 20, 7139–7152, 2020. Ault et al. used Nd:YAG lasers at 266 nm, a wavelength more comparable to the KrF excimer laser than to the ArF laser.

We have amended the text on line 121 to include the following text from your suggestion:

"The strong $NO^+$ signals observed in this work may be a result of using a 193 nm excimer laser, which is more effective at ionizing nitrogen containing species compared to the 248 nm (Passig et al., 2021) and 266 nm laser (Ault et al., 2010) used in the other studies, which are more sensitive to iron and transition metals due to a resonance effect. Finally, this could also be indicative of atmospheric aging processes."

- lines 159-162: Anthropogenic particles in the respective size mode are subject to long-range transport. So, why exclude particles detected over land and in polar regions? While shipping activity is much lower in the polar regions, the particles there are important for climate impacts, e.g. through deposition on ice and albedo changes. Were there not enough V-containing particles detected? The exclusion of these areas should be better justified.

Almost all the ATom data is over the ocean, and the very few flights over land (for example, transits across northern Alaska) were not representative of larger areas. The polar regions are a different category than the open oceans. Not only is there essentially no shipping, the boundary layer and wind transport are different over ice than over open ocean. Finally, near Antarctica the flight patterns were different. It would considerably complicate the manuscript to include the polar regions. We have added a brief mention of the different boundary layer to the caption of Figure 4.

- lines 163-170: The conservative determination of V particles from shipping is appropriate. An important consideration is the ageing of these particles. Both an increase and a decrease in the organic carbon content of the particles is possible during ageing and would have a direct effect on particle identification using the criteria of V+/VO+ peak intensity relative to neighbouring peaks. This is discussed later in the manuscript, but may be briefly mentioned already here.

The criteria of V+/VO+ peak intensity relative to neighbouring peaks, is to distinguish vanadium from organic peaks. Although an enhancement in organics could affect this interpretation, the organic content increase would have to be significant, which we do not expect to see during the atmospheric aging of vanadium-containing ship exhaust particles. We have added the near on line 179 in the manuscript:

"In addition, vanadium signals could be obscured by larger organic signals as particles age, although the organic peaks do not get so large (section 3.5) for this to noticeably affect classification."

- General comment: I am missing the total number of V-particles in the respective measurements/flights. I may have overlooked it, but this information would give an estimate of the statistical quality.

This is now stated near line 210 during the discussion of Figure 4:

"Each ~12° latitude bin in Figure 4 includes about 1000 to 4000 particles in the MBL and about 4000 to >10000 particles above the MBL during each deployment. With about 1% of particles in the MBL containing vanadium and less than that above the MBL, each point in **Fig. 4** represents tens of vanadium-containing particles."

- lines 201-214: Could the authors provide an example of these diluted plumes? For example as a plot in the supplement?

We have added Figure S5.

- Figure 5: Again, I am missing absolute particle numbers. If the feature in the tropical Pacific is attributed to a single ship plume, as stated in the main text, I assume the particle numbers to be relatively low. Since sampling time is limited and particle concentration in the free remote atmosphere is low, low particle numbers are an inherent problem of such measurements and not a drawback of the study.

Particle numbers are now stated near line 210 during the discussion of Figure 4:

- line 255: Did you really observe a sufficient signal of oxalate in positive mode? I cannot see this signal in the mass spectra (Fig. 2).

Oxalate is indeed difficult to see in positive mode. We have removed mentioning it here.

Finally, we apologize for the slow response to reviewers. The lead author started a new job during the review process, which led to some delays in our response times.